# MEMECAP: A Dataset for Captioning and Interpreting Memes

**EunJeong Hwang**[1,2] **and Vered Shwartz**[1,2]
[1] University of British Columbia  [2] Vector Institute for AI
{ejhwang,vshwartz}@cs.ubc.ca

## Abstract

Memes are a widely popular tool for web users to express their thoughts using visual metaphors. Understanding memes requires recognizing and interpreting visual metaphors with respect to the text inside or around the meme, often while employing background knowledge and reasoning abilities. We present the task of meme captioning and release a new dataset, MEMECAP. Our dataset contains 6.3K memes along with the title of the post containing the meme, the meme captions, the literal image caption, and the visual metaphors. Despite the recent success of vision and language (VL) models on tasks such as image captioning and visual question answering, our extensive experiments using state-of-the-art VL models show that they still struggle with visual metaphors, and perform substantially worse than humans.

## 1 Introduction

Web users frequently communicate their thoughts and feelings online using memes (Buchel, 2012; Tanaka et al., 2022). Memes are created by taking an existing widespread image and attaching new meaning to it by altering the text inside the image. For example, in Figure 1, Tom cat is a metaphor for the person who posted the meme and the cats he is shaking hands with represent his two regular followers who always like his posts. This incongruity between the image and the text makes memes humorous (Tanaka et al., 2022).

Because of their complementary nature, interpreting the meaning of a meme requires understanding both the visual and text modalities. Moreover, memes are often posted on social media platforms along with additional text, such as "one of them is my alt" in Fig. 1, which is further needed to understand the meme.

Recently, there is a surge of vision and language (VL) models (e.g. Alayrac et al., 2022; Li et al., 2023; OpenAI, 2023). VL models have shown remarkable capabilities in generating detailed and

**Title:** one of them is my alt

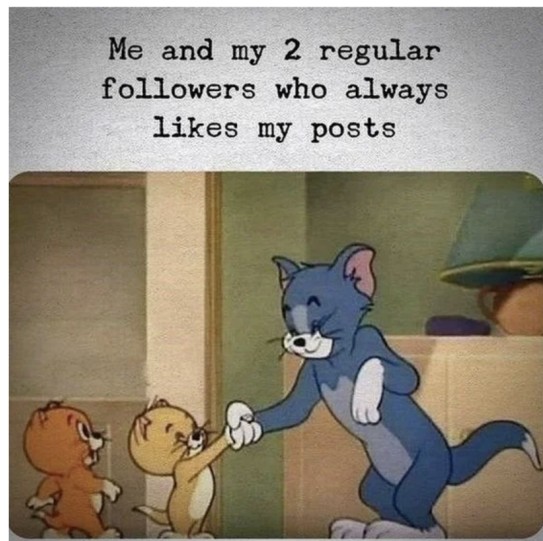

**Caption:** Meme poster appreciates their only two followers and one of them is their alternative account.

Figure 1: A meme and its title. The caption describes what the meme poster was trying to convey.

accurate descriptions of images in both zero-shot and in-context setups. Such models are first pre-trained on language-only and vision-only datasets, and then trained on tasks such as image captioning and visual question answering, where the redundancy between the vision and language is used to embed them in a shared space. For example, the majority of image captions in existing datasets describe what is depicted in the image, at most adding subjective interpretations or inferences about the story behind the image (Alikhani et al., 2020). In contrast, there is little work on visual metaphors to date (Zhang et al., 2021; Chakrabarty et al., 2023).

In this paper, we are investigating whether VL models can successfully interpret memes. We propose the task of *meme captioning*, in which models are presented with a meme along with its title (e.g. the title of the post containing the meme), and is tasked with generating a concise caption describing

the meaning of the meme. This task goes beyond object recognition and language understanding. It is challenging due to the metaphorical role of the visual content of the meme (Scott, 2021). For example, in Fig. 1, the model needs to recognize that Tom cat is merely a metaphor for the meme poster, and that handshaking signals appreciation. The literal content of the image, such as Tom or the handshake, should not be part of the meme caption. Recognizing and interpreting such metaphors involve detecting facial expressions, the tone expressed in the texts, making commonsense inferences, and more (Bitton-Guetta et al., 2023).

To that end, we collected a meme captioning dataset MEMECAP, containing 6,384 memes along with their captions. Each meme is also annotated with the literal image description (e.g. "Tom cat is shaking hands with two small cats and smiling"), and the visual metaphors (e.g. Tom is a metaphor for the meme poster).

We establish comprehensive baseline performances with recent large-scale VL models, in various training setups (e.g. zero-shot, few-shot, finetuning), and inputs (i.e. meme, title, literal image captions, and metaphors). Human evaluation of the generated captions shows that models are far from humans in captioning memes. In particular, models tend to ignore important visual or textual elements, and instead, repeat the text inside the meme or make up fake elements. Our findings merit future research on this task. [1]

## 2 Background

### 2.1 Metaphors

Most work on metaphors is limited to textual metaphors, and pertains to collecting resources (Dodge et al., 2015), detecting or interpreting metaphorical expressions in context (Choi et al., 2021; Chakrabarty et al., 2021a; Aghazadeh et al., 2022; Chakrabarty et al., 2022), and metaphor generation (Stowe et al., 2021; Chakrabarty et al., 2021b).

Recently, there has been interest in visual metaphors. Visual metaphors occur when a target concept is compared to another visual element (vehicle) (Forceville, 1996). MultiMET (Zhang et al., 2021) and Met-Meme (Xu et al., 2022) are two datasets of text-image pairs with annotations for the existence and types of metaphors, sentiment,

and more. Chakrabarty et al. (2023) tested image generation models on prompts involving a visual metaphor such as "My bedroom is a pigsty". They found the unsatisfactory performance can be improved by using a large language model (LLM) to interpret the visual metaphors and add details to the prompt, such as "messy bedroom". Akula et al. (MetaCLUE; 2023) introduces a set of tasks pertaining to visual metaphors in synthetic images, such as retrieval and captioning.

Finally, the Image Recognition of Figurative Language dataset (IRFL; Yosef et al., 2023) presents an idiom, metaphor, or simile, along with 4 images, with the goal of selecting the image that matches the figurative expression. The distractors consist of an image that depicts the expression literally, for example a picture of a cheetah for the simile "as fast as a cheetah". This dataset is challenging for state-of-the-art VL models.

### 2.2 Memes

Recent work on memes focused on detecting hateful or harmful content in memes (Kiela et al., 2021; Qu et al., 2022; Sharma et al., 2023), classifying memes to humorous or not (Tanaka et al., 2022), and analyzing the sentiment of memes (Sharma et al., 2020). Earlier work automatically generated a text to go inside the meme (Wang and Wen, 2015), and the memes themselves (e.g. the ImgFlip575K dataset).[2]

Although MultiMET (Zhang et al., 2021) does not focus specifically on memes, the images were collected from a range of sources including social media, which contains memes. The similar Met-Meme dataset (Xu et al., 2022) focuses on memes. Differently from our work, both datasets contain annotations for visual metaphors while MEMECAP also contains meme captions.

### 2.3 Other Image Datasets

The WHOOPS benchmark (Bitton-Guetta et al., 2023) consists of unconventional human-created and machine-generated images that defy commonsense (e.g. an image of "Albert Einstein holding a smartphone"), along with their textual descriptions. It's meant to be used for image captioning, image-text matching, visual question answering, and explanation generation. In contrast, our work focuses on memes, and tests models on their ability to interpret real memes posted by web users.

---

[1] Our code and data are available at:
https://github.com/eujhwang/meme-cap

[2] https://github.com/schesa/ImgFlip575K_Dataset

| | #Memes | #M-Cap | #I-Cap | #Mph |
|---|---|---|---|---|
| Train+Val | 5,828 | 1.0 | 1.0 | 2.1 |
| Test | 559 | 3.4 | 1.0 | 3.1 |

Table 1: The number of memes in MEMECAP, and the average number of meme captions (M-Cap.), image captions (I-Cap.), and metaphorical keywords (Mph) per meme.

Another multi-modal benchmark which is similar to ours is the New Yorker Cartoon Caption Contest (Hessel et al., 2023). This benchmark involves 3 tasks: matching a caption to a cartoon, evaluating the quality of the caption, and explaining the joke. While both memes and cartoons use a combination of visual and textual elements to convey humor, memes are based on recognizable images that are modified and repurposed to create new meanings based on shared cultural knowledge. Cartoons, on the other hand, are originally drawn illustrations, often in the form of comic strips, that convey a more complex narrative. Further, while Hessel et al. (2023) focus on discriminative matching (i.e. selecting the more appropriate caption) and generating an explanation, in this paper we present a generative task, i.e. generating a caption to describe a meme.

## 3 The MEMECAP Dataset

The overall data collection and annotation process is illustrated in Figure 2. We collected memes (Sec 3.1) and crowdsourced their captions (Sec 3.2). We present the data splits and statistics in Sec 3.3.

### 3.1 Memes

We scraped memes from Reddit using the publicly available API.[3] In particular, we focused on the subreddit /r/memes and collected posts that contained a meme with a post title. To ensure that the text and image are complementary, we manually examined the memes and excluded memes that lacked any text or contained an excessive number of characters. To exclude offensive content from the dataset, we filtered out memes with profanity in the text using the Google banned word list.[4] We also filtered out images with sexual content, for which the NudeNet Classifier returned an unsafe score higher than 0.9.[5]

### 3.2 Captions

We conducted two rounds of annotations to obtain the captions. In the first round, we collected the literal image descriptions, disregarding the text in the memes, while in the second round, we collected the meme caption along with the visual metaphors.

**Literal Image Captions.** We asked workers to caption the image, disregarding the text. For example, a suitable literal image caption for Figure 1 is "Tom cat is shaking hands with two small cats and smiling". To prevent biasing the workers with the text inside the meme, we identified and removed the text in the meme using the LaMa inpainting tool (Suvorov et al., 2021). We collected one caption for each meme, which we manually verified.

**Meme Captions.** We showed a second set of annotators the full meme, title, and literal image caption, and asked them to provide a meme caption. This HIT included two steps. First, workers were asked to indicate for each term in the literal image caption whether it was used metaphorically, and if so, what was the target of the metaphor (e.g., "Tom cat" is a metaphor for the meme poster). We then instructed the workers to write a concise caption describing the meaning that the meme poster was trying to convey, while excluding the metaphor vehicles (e.g., not mentioning Tom). We collected one caption for each meme in the training set, and 2 to 4 captions for memes in the test set.

Both rounds of annotations were conducted on Amazon Mechanical Turk (MTurk). To ensure the quality of annotations, we required that workers were located in English-speaking countries (e.g. US, UK, Canada, Australia, and New Zealand), had an acceptance rate of at least 98% on 5,000 prior HITs, and passed a qualification test similar to the task.

We excluded from the dataset any memes that workers in each of the rounds marked as offensive, sexual, hateful, or uninterpretable.

### 3.3 Final Dataset

We clustered the examples in the dataset based on the vector representation of their meme captions using OPT2.7b (Zhang et al., 2022). To ensure the diversity of topics in both the training and test sets, we then sampled 10% of the memes from each cluster and assigned them to the test set, and the rest of the memes into the training and validation

---

[3] https://www.reddit.com/dev/api/
[4] https://github.com/coffee-and-fun/google-profanity-words
[5] https://github.com/notAI-tech/NudeNet

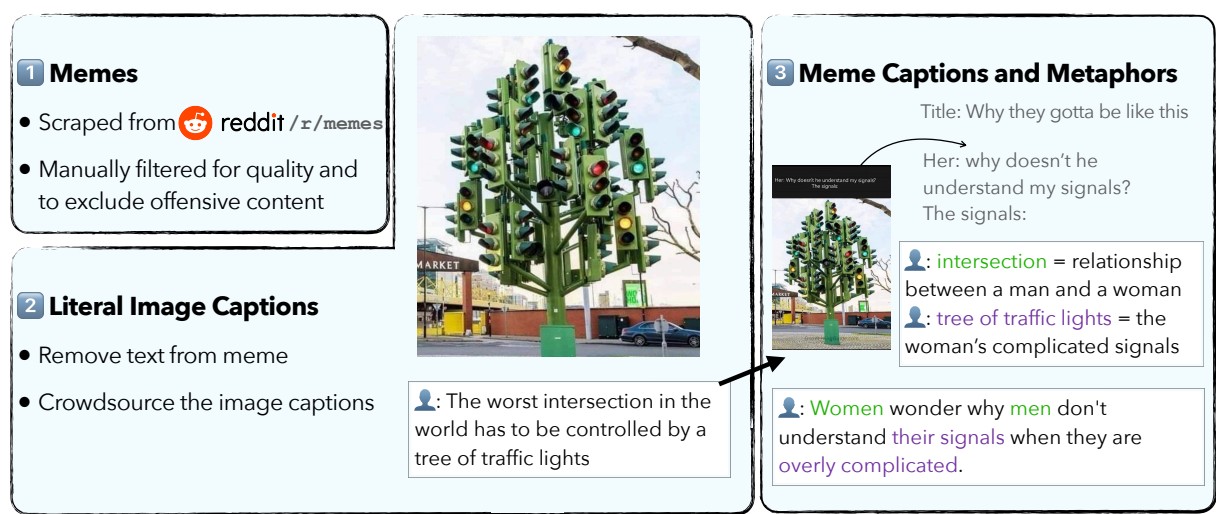

Figure 2: Overall process of collecting memes, literal image captions, visual metaphors, and meme captions.

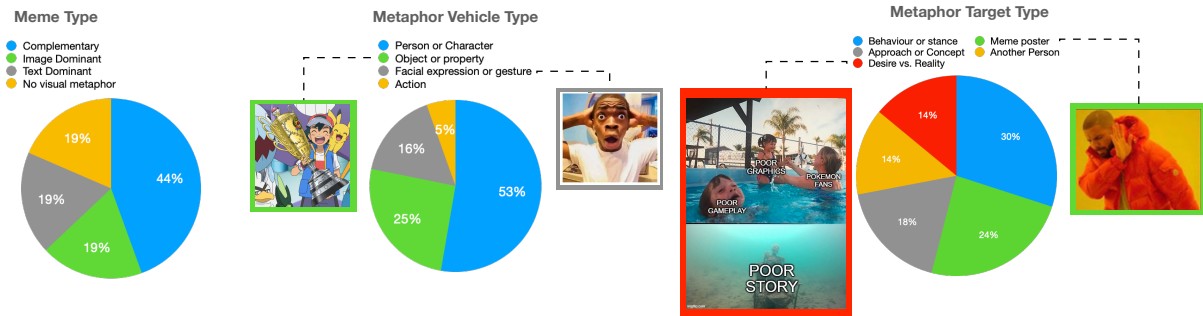

Figure 3: (1) **Meme Type**: Percent of memes with no visual metaphors, and with metaphors that can be understood with the text alone, vision alone, or both (complementary). (2) **Metaphor Vehicle Type**: Types of visual elements used to convey a metaphorical meaning. (3) **Metaphor Target Type**: The intended meanings of the metaphors.

set.[6] Table 1 shows the statistics of our dataset.

### 3.4 Types of Metaphors

We manually analyzed 28 memes along with their metaphor annotations.

**Meme Type.** First, following Zhang et al. (2021) and Xu et al. (2022), we categorized the memes into three categories: *text dominant* and *image dominant*, where the text or the image respectively may be enough to understand the metaphor, and *complementary*, where both modalities are required. We added a fourth category for memes that had no metaphor, i.e. whose meaning is conveyed explicitly in the text. The left part of Figure 3 shows that the 44% of memes are complementary, but each of the other categories is also prominent with 19%.

We then looked at the human annotations we obtained in Sec 3.2 for the metaphors in each meme. We looked at the vehicle, i.e. the visual element

used to convey the metaphorical meaning, as well as the target, i.e. the meaning itself.

**Metaphor Vehicle Type.** The middle part of Fig 3 shows that the most common vehicle is a person or a character, followed by objects (such as the trophy), facial expressions or gestures (such as the surprised look on the man's face), and actions.

**Metaphor Target Type.** The types of targets are displayed in the right part of Fig 3. The majority of the metaphors describe either a behavior or stance towards a certain topic, or the meme poster themselves (with a person vehicle, such as Drake). Other categories are an approach or a concept (for which the meme poster expresses a certain stance), another person, and a "desire vs. reality" meme such as the drowning meme illustrated in Fig 3.

### 4 Experimental Setup

We report the performance of various baselines on MEMECAP. All models are tasked with generating

---

[6]Note that our dataset doesn't contain duplicate memes.

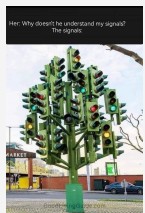

Figure 4: An example of the few-shot setup with the following inputs: meme, image description, and the text inside the meme. The figure shows the last in-context meme and the target meme.

## 4.1 Models

We experiment with two state-of-the-art VL models that can generate text conditioned on both text and images, as well as one language model.

**Open Flamingo.** Flamingo was initialized with a pre-trained LLM and a pre-trained vision model, and further trained on vision and language tasks, keeping the pre-trained models frozen. The interaction between the two modalities is facilitated with a gated cross-attention dense block. Since the original model is not publicly available, we use the open version, OpenFlamingo-9B (Awadalla et al., 2023). OpenFlamingo is built on top of LLaMA 7B (Touvron et al., 2023) and CLIP ViT/L-14 (Radford et al., 2021), and was trained on 5M samples from the Multimodal C4 dataset (Zhu et al., 2023b) and 10M samples from LAION-2B (Schuhmann et al., 2022).

**MiniGPT4.** MiniGPT4 (Zhu et al., 2023a) is similarly composed of frozen pre-trained language and vision models, and it employs a single projection layer to align the visual and language features. Since GPT4's architecture and training data remain a mystery, we utilize MiniGPT4 as an alternative to GPT4 (OpenAI, 2023).[7] It has similar capabilities to GPT-4 in understanding and generating the context (Zhu et al., 2023a). For its language model, MiniGPT4 uses Vicuna (Chiang et al., 2023), which is built on top of LLaMA-13B and performs on par with ChatGPT (OpenAI, 2023). For its vision component, it uses BLIP-2 (Li et al., 2023), which consists of CLIP ViT-G/14 and a Q-Former architecture. MiniGPT4 was trained on various multimodal datasets, including images

from LAION (Schuhmann et al., 2022), Conceptual Captions (Sharma et al., 2018), and SBU (Ordonez et al., 2011).

**LLaMA** LLaMA (Touvron et al., 2023) is a transformer-based language model that was trained on trillions of tokens from exclusively publicly-available data. The LLaMA-13B model outperforms GPT-3 (Brown et al., 2020) on most benchmarks. We use the LLaMA-7B model, which achieves comparable performance to the LLaMA-13B model on most benchmarks. Since LLaMA is a language model rather than a VL model, its access to the visual content is through the image caption and the OCR text alone.

## 4.2 Evaluation Setup

**Inputs.** We test the models with different input settings. In the setup which is the most comparable to humans, we provide the models with the meme and title. We also experiment with setups that aid the model. One such input is the image caption, which can help the model focus on the language modality and ignore the image. The second such input is the text inside the meme, that we extracted using EasyOCR,[8] which helps the model focus on the visual aspects of the image and includes the text inside the image as part of the language input. We incrementally added each of these inputs.

**Learning Setups.** We evaluate all models in a zero-shot setup. Flamingo and LLaMA enable in-context learning, so we experiment with 4, 8, and 12 shots. An example prompt (including the meme, title, image caption, and text inside the meme) is illustrated in Figure 4. MiniGPT4 works in a chat format, so rather than in-context learning, we use it in either a zero-shot setup, or fine-tuned on our training set.

Lastly, motivated by Chakrabarty et al. (2023) and Zhang et al. (2023), we also tested models in a

---

[7]The version of GPT-4 available through the OpenAI API doesn't support images.

[8]https://github.com/JaidedAI/EasyOCR

Chain of Thought (CoT) style prompting (Wei et al., 2022). In our case, we elicit multi-step reasoning from the LLM by providing the visual metaphors, using the following prompt:

```
<image>This is a meme with the title "{title}".
The image description is "{image caption}".
The following text is written inside the meme:
"{OCR text}".
What is the meme poster trying to convey?
Rationale: "{keyword1}" is a metaphor for
"{meaning1}". "{keyword2}" is a metaphor for
"{meaning2}".
Answer:
```

## 5 Results

We evaluated the performance of the various models with both automatic metrics (Sec 5.1) and human evaluation (Sec 5.2). We show that the vision and language modalities are complementary through ablation tests (Sec 5.3).

### 5.1 Automatic Evaluation

To evaluate the quality of the generated captions, we use standard metrics for automatic evaluation of generative tasks: BLEU (Papineni et al., 2002) ROUGE (Lin, 2004), and BERTScore (Zhang et al., 2020) (using microsoft/deberta-xlarge-mnli). BLEU and ROUGE are based on n-gram overlap between the generated captions and human-written reference captions, while BERTScore measures the semantic similarities between the two.

Table 2 shows the performance of the various models and input setups in terms of these metrics. For the few-shot setup, we show the best performance across (4, 8, and 12 shots). See Appendix A for the full results.

**Models.** Flamingo dominates MiniGPT4 across all metrics, with a gap of 15, 12, and 6 points in BLEU, ROUGE, and BertScore respectively for the best setups. This is likely due to the lengthy captions generated by MiniGPT4, despite the prompt including the instruction to generate a single sentence. Finally, the LLaMA model is highly competitive with Flamingo despite not having access to the image itself. It appears that the image captions and OCR text provide sufficient information.

**Learning Setups.** The Flamingo performance significantly improves from the zero-shot to few-shot setting, and continues to improve from 4 to 8 shots but slightly decreases at 12 shots (see Appendix A). MiniGPT4 achieved better performance in the zero-shot setup, while fine-tuning its last layer significantly decrease the performance. As we show in Sec 5.2, while the fine-tuned model learns to generate short captions, it tends to hallucinate more. We hypothesize that fine-tuning only the last layer is ineffective.

**Inputs.** In the few-shot setups, the best performance is achieved with as many of the inputs as possible, i.e. including both the image caption and the OCR text, despite the redundancy with the visual inputs. This might be due to suboptimal cross-modal interaction in VL models. While prior work showed that explicitly stating the metaphors helps image generation models generate better images (Chakrabarty et al., 2023), we did not see a similar gain in meme captioning.

### 5.2 Human Evaluation

We focused on the models with the full set of inputs except for the rationales (meme+title+img cap+OCR text) and evaluated the performance of all models (focusing on 4-shots for the few-shot setups), with respect to the following criteria:

- **Correctness**: Does the caption correctly convey the meaning the meme poster wanted to convey?

- **Appropriate Length**: Is the caption length appropriate for conveying the meaning (i.e. it is not too verbose)?

- **Visual Completeness**: Does the caption describe all the important elements in the image?

- **Textual Completeness**: Does the caption describe all the important elements in the text inside the meme and the title text?

- **Faithfulness**: Are all the elements of the caption supported by either the visual or text elements (i.e. there are no made-up elements)?

We randomly sampled 30 memes along with their model-generated and human-written captions. The annotation was performed by students in the lab, and we took the majority vote across 3 annotators. Figure 5 shows the performance according to the human evaluation. All models perform significantly worse than humans, except for appropriate length criteria, with 36.6, 29.3, 24.5, and 18.4 point differences on correctness, textual completeness, visual completeness, and faithfulness respectively.

| Model | Setup | Inputs | BLEU-4 | ROUGE-L | BERT-F1 |
|---|---|---|---|---|---|
| **Flamingo** | **zero-shot** | meme+title | 19.36 | 31.51 | 65.69 |
| | | meme+img cap | 16.10 | 29.08 | 64.71 |
| | | meme+title+img cap | 19.61 | 30.92 | 65.51 |
| | | meme+title+img cap+OCR text | 19.31 | 32.51 | 66.84 |
| | **zero-shot CoT** | meme+title+img cap+OCR text+rationale | 2.49 | 15.89 | 58.23 |
| | **few-shot** | meme+title | 25.89 | 39.41 | 70.83 |
| | | meme+img cap | 26.96 | 39.53 | 70.91 |
| | | meme+title+img cap | 26.44 | 39.42 | 71.04 |
| | | meme+title+img cap+OCR text | 26.73 | **43.47** | 73.86 |
| | **few-shot CoT** | meme+title+img cap+OCR text+rationale | **27.02** | 43.46 | 74.32 |
| **MiniGPT4** | **zero-shot** | meme | 06.17 | 22.20 | 63.31 |
| | | meme+title | 14.37 | 30.70 | 66.19 |
| | | meme+img cap | 10.36 | 26.22 | 64.39 |
| | | meme+title+img cap | 12.49 | 28.51 | 65.81 |
| | | meme+title+img cap+OCR text | 12.46 | 31.44 | 68.62 |
| | **zero-shot CoT** | meme+title+img cap+OCR text+rationale | 12.57 | 31.70 | 68.45 |
| | **fine-tuned** | meme+title+img cap+OCR text | 7.50 | 27.88 | 65.47 |
| | **fine-tuned CoT** | meme+title+img cap+OCR text+rationale | 7.25 | 26.68 | 65.86 |
| **LLaMA** | **zero-shot** | title+img cap | 19.72 | 31.42 | 66.38 |
| | | title+img cap+OCR text | 20.77 | 36.48 | 69.67 |
| | **zero-shot CoT** | title+img cap+OCR text+rationale | 6.72 | 20.56 | 61.38 |
| | **few-shot** | title+img cap | 26.41 | 38.70 | 70.01 |
| | | title+img cap+OCR text | 26.63 | 43.41 | **74.71** |
| | **few-shot CoT** | title+img cap+OCR text+rationale | 26.40 | 42.95 | 74.00 |

Table 2: Performance in terms of automatic metrics of the various models and learning setups (with 4 shots for the few-shot setup). We report the full experimental results, including 8 shots and 12 shots, in Appendix A.

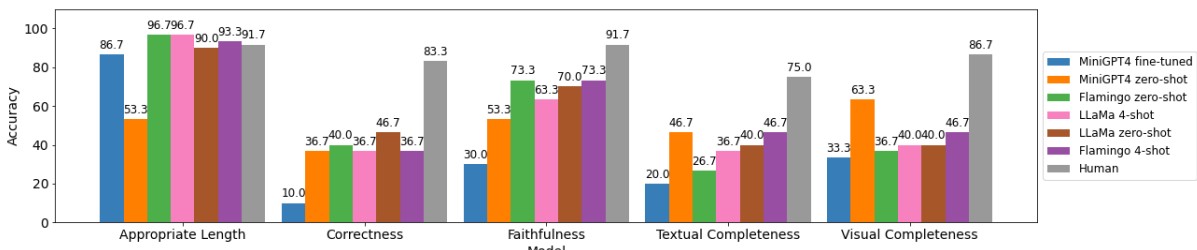

Figure 5: Performance in terms of human evaluation.

**Models.** Model performances differ by criteria. Flamingo and LLaMA are more correct and faithful, while MiniGPT4 is more visually complete.

**Learning Setups.** For Flamingo, the few-shot models improve in textual and visual completeness upon the zero-shot model, but not in terms of correctness and faithfulness. This may suggest that while access to examples improves the model's understanding of the task, it might also confuse it with information irrelevant to the target meme. LLaMA doesn't gain any performance improvements from in-context examples, likely for the same reason. Without the visual features, it might struggle even more to separate the text (title, image caption, and OCR) of the different examples.

MiniGPT4 zero-shot is very verbose, but the fine-tuned model learns to output captions in the length of its training examples. Unfortunately, these captions are far worse than those of the zero-shot model in all criteria. The zero-shot version generates verbose captions that include a lot of information, often conveying the correct meaning along with irrelevant information such as literal descriptions of the image. Conversely, the fine-tuned version adapts to the "correct" length but it often fails to focus on the relevant parts, leading to incorrect or incomplete captions. We hypothesize that the frozen language and vision model may not have enough information about interpreting memes, and simply fine-tuning the last projection layer of the model is not enough to produce high-quality captions. This conclusion is consistent with Zhou et al. (2023), according to which most knowledge in LLM is learned during the pre-training stage.

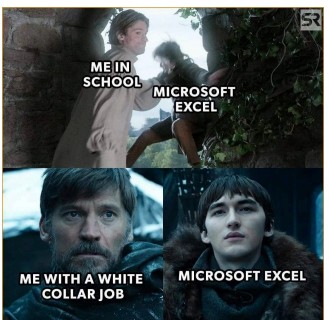

**Error: unfaithful**

> **Title**: This is my character arc
>
> **Image caption**: This is a poster of Game of throne from the tower scene.
>
> **Human-written meme caption**: Meme poster abandoned Microsoft Excel in school, but need to use it after they get their white collar job.
>
> **Model-generated meme caption**: Meme poster is trying to convey that they want to be successful in life.

**Error: visually incomplete (copying the text inside the meme)**

> **Title**: Based on a true story
>
> **Image caption**: Spongebob is eagerly watching TV
>
> **Human-written meme caption**: Meme poster finds it entertaining to read through long comment threads of arguments that happened in the past.
>
> **Model-generated meme caption**: Meme poster is trying to convey that they read a 153 comment long argument that happened 7 years ago.

Figure 6: Examples of incorrect meme captions generated by the few-shot Flamingo model.

**Common Errors.** Figure 6 shows two examples of meme captions generated by Flamingo 4-shot along with the types of errors they exhibit. The top example demonstrates an unfaithful caption because neither the meme nor the title conveys anything about being successful in life. The bottom example illustrates a common error in which the model copies text from inside the meme while ignoring important visual elements. In this case, Spongebob's smile indicates the meme poster's positive attitude towards reading old and long forum threads, but the model-generated caption misses it. Another common error (not illustrated here) occurs when the model treats visual elements too literally, failing to interpret the metaphor. Finally, in some cases, the model might lack sufficient background knowledge to correctly interpret the meme.

### 5.3 Ablation Tests

The analysis in Sec 3.4 shows that interpreting most memes in MEMECAP will require understanding both the visual and text modalities. We are interested in the extent that models make use of each modality. To that end, we perform an ablation test to exclude each modality. Table 3 presents the results in terms of automatic metrics.

In most cases, the best performance is achieved with both modalities. For Flamingo (zero-shot and few-shot), excluding the meme results in more decrease in performance than excluding the title, in-

| Model | k | Inputs | ΔBL | ΔRG | ΔBT |
|---|---|---|---|---|---|
| **Flamingo** | **0** | full | 19.36 | 31.51 | 65.69 |
| | | -title | -2.29 | -1.35 | -0.6 |
| | | -meme | -1.49 | -1.93 | -1.71 |
| | **4** | full | 25.89 | 39.41 | 70.83 |
| | | -title | +0.35 | +0.12 | -0.19 |
| | | -meme | -0.14 | -0.85 | -1.86 |
| **MiniGPT4** | **0** | full | 14.37 | 30.70 | 66.19 |
| | | -title | -8.2 | -8.5 | -2.88 |
| | | -meme | +3.5 | -1.12 | -2.21 |
| **LLaMA** | **0** | full | 19.72 | 31.42 | 66.38 |
| | | -title | -0.88 | -0.93 | -0.62 |
| | | -img cap | -1.85 | -1.84 | -2.4 |
| | **4** | full | 26.41 | 38.70 | 70.01 |
| | | -title | -0.69 | -0.73 | -0.67 |
| | | -img cap | -0.66 | -0.14 | -1.04 |

Table 3: Comparison models with both language and visual inputs (title+ima cap for LLaMA, title+meme for VL models), compared to one modality. BL = BLEU, RG = ROUGE, BT = BERT. k = number of shots.

dicating that the model relies more on the visual modality than the information provided by the title. The same is true for LLaMA (in both settings), for which excluding the image caption yields worse performance. This is expected since the title is typically secondary in informativeness to the meme. In addition, Flamingo still has access to the text inside the meme via visual features.

Conversely, MiniGPT4 exhibits a higher dependency on textual modality, resulting in a significant decrease when the title is not provided. Since

MiniGPT4 shows higher textual and visual completeness when the OCR text is provided (§5.2), we hypothesize that MiniGPT4 makes limited usage of the visual modality.

# 6 Conclusion

We present MEMECAP, the first meme captioning dataset. MEMECAP is challenging for the existing VL models, as it requires recognizing and interpreting visual metaphors, and ignoring the literal visual elements. The experimental results using state-of-the-art VL models indeed show that such models are still far from human performance. In particular, they tend to treat visual elements too literally and copy text from inside the meme. Our work opens up interesting future research on recognizing visual metaphors, interpreting them with respect to a textual context, and generating meme captions that are complete with respect to both modalities without creating fake elements.

## Limitations

**Quality of Metaphor Annotations.** We put our best efforts into manually verifying the collected data, and indeed the human performance in Section 5.2 shows the human-written captions are of high quality. With that said, we noticed that the quality of the visual metaphors is inconsistent. We believe that while people are capable of explaining a meme, they don't always know to map the visual vehicles into textual targets. This likely explains why adding the metaphors as inputs didn't improve the performance.

**Subjectivity and Background Knowledge.** The meme captioning task involves employing background knowledge which may vary between annotators. To that end, we manually checked the meme captions to minimize the number of incorrect captions in the dataset. In addition, there is some level of subjectivity with respect to the evaluation criteria for the meme caption quality. For this reason, we ensured a high quality of annotations by having in-house annotators that could ask clarification questions, but some subjectivity still remains.

## Ethics Statement

**Data** All the datasets used in our work are publicly available. Our dataset is collected from Reddit and may contain offensive, hateful, or sexual content. Despite our best efforts to filter them out as described in Section 3, we found people have different criteria for what they perceive as offensive, hateful, or sexual, and thus, such content may still exist in our data.

**Data Collection** We use Amazon Mechanical Turk to collect 6.3K image descriptions and 7.7K meme captions. We paid $0.03 for the image captioning task and $0.16 for the meme captioning task. The annotators were compensated with an average hourly wage of $13, which is comparable to the US minimum wage. We did not collect any personal information from annotators.

**Models** Our dataset may include some offensive content or mild expletives and this can amplify potentially biased and unethical answers. In addition, the large pre-trained VL models we used for the experiments are trained on a large-scale publicly available web corpus and may also bring some bias when generating sentences.

## Acknowledgements

This work was funded, in part, by the Vector Institute for AI, Canada CIFAR AI Chairs program, an NSERC discovery grant, and a research gift from AI2.

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

# A  Additional Experimental Results

We show the full experimental results in Table 4.

| Model | # Shots | Input | BLEU-4 | ROUGE-L | BERT-F1 |
|---|---|---|---|---|---|
| **Flamingo** | **0-shot** | meme | 17.07 | 30.16 | 65.09 |
| | | meme+title | 19.36 | 31.51 | 65.69 |
| | | meme+img cap | 16.10 | 29.08 | 64.71 |
| | | meme+title+img cap | 19.61 | 30.92 | 65.51 |
| | | meme+title+img cap+OCR text | 19.31 | 32.51 | 66.84 |
| | **0-shot CoT** | meme+title+img cap+OCR text+rationale | 2.49 | 15.89 | 58.23 |
| | **4-shot** | meme | 26.24 | 39.53 | 70.62 |
| | | meme+title | 25.89 | 39.41 | 70.83 |
| | | meme+img cap | 26.96 | 39.53 | 70.91 |
| | | meme+title+img cap | 26.44 | 39.42 | 71.04 |
| | | meme+title+img cap+OCR text | 26.73 | 43.47 | 73.86 |
| | **4-shot CoT** | meme+title+img cap+OCR text+rationale | 27.02 | 43.46 | 74.32 |
| | **8-shot** | meme | 27.38 | 39.96 | 70.92 |
| | | meme+title | 26.99 | 40.00 | 71.26 |
| | | meme+img cap | 28.11 | 40.32 | 71.24 |
| | | meme+title+img cap | 27.30 | 40.00 | 71.32 |
| | | meme+title+img cap+OCR text | 28.70 | 43.54 | 74.33 |
| | **8-shot CoT** | meme+title+img cap+OCR text+rationale | - | - | - |
| | **12-shot** | meme | 26.74 | 38.89 | 70.20 |
| | | meme+title | 27.32 | 40.13 | 70.86 |
| | | meme+img cap | 26.63 | 39.24 | 70.49 |
| | | meme+title+img cap | 27.09 | 39.60 | 70.48 |
| | | meme+title+img cap+OCR text | - | - | - |
| | **12-shot CoT** | meme+title+img cap+OCR text+rationale | - | - | - |
| **LLaMA** | **0-shot** | title | 17.87 | 29.58 | 63.98 |
| | | img cap | 18.84 | 30.49 | 65.76 |
| | | title+img cap | 19.72 | 31.42 | 66.38 |
| | | title+img cap+OCR text | 20.77 | 36.48 | 69.67 |
| | **0-shot CoT** | title+img cap+OCR text+rationale | 6.72 | 20.56 | 61.38 |
| | **4-shot** | title | 25.75 | 38.56 | 68.97 |
| | | img cap | 25.72 | 37.97 | 69.34 |
| | | title+img cap | 26.41 | 38.70 | 70.01 |
| | | title+img cap+OCR text | 26.63 | 43.41 | 74.71 |
| | **4-shot CoT** | title+img cap+OCR text+rationale | 26.40 | 42.95 | 74.00 |
| | **8-shot** | title | 27.18 | 39.19 | 69.66 |
| | | img cap | 27.25 | 38.61 | 69.67 |
| | | title+img cap | 27.99 | 39.69 | 70.76 |
| | | title+img cap+OCR text | 28.80 | 44.10 | 74.71 |
| | **8-shot CoT** | title+img cap+OCR text+rationale | 26.32 | 42.06 | 73.95 |
| | **12-shot** | title | 25.71 | 37.15 | 68.26 |
| | | img cap | 25.65 | 36.37 | 68.65 |
| | | title+img cap | 26.63 | 38.57 | 69.96 |
| | | title+img cap+OCR text | 28.76 | 43.18 | 73.96 |
| | **12-shot CoT** | title+img cap+OCR text+rationale | - | - | - |
| **MiniGPT4** | **0-shot** | meme | 06.17 | 22.20 | 63.31 |
| | | meme+title | 14.37 | 30.70 | 66.19 |
| | | meme+img cap | 10.36 | 26.22 | 64.39 |
| | | meme+title+img cap | 12.49 | 28.51 | 65.81 |
| | | meme+title+img cap+OCR text | 12.46 | 31.44 | 68.62 |
| | **0-shot CoT** | meme+title+img cap+OCR text+rationale | 12.57 | 31.70 | 68.45 |
| | **finetuned** | meme+title+img cap+OCR text | 7.50 | 27.88 | 65.47 |
| | **FT CoT** | meme+title+img cap+OCR text+rationale | 7.25 | 26.68 | 65.86 |

Table 4: 0, 4, 8, 12 shot results with Flamingo, LLaMA, and MiniGPT4 models. "-" indicates the model ran out of memory.