# OpenReview forum: "MemeCap: A Dataset for Captioning and Interpreting Memes"
_EMNLP/2023/Conference — EMNLP 2023 Main_

### Official Review · Reviewer_XU7o · 2023-08-03

**Soundness:** 3

**Excitement:**

4: Strong: This paper deepens the understanding of some phenomenon or lowers the barriers to an existing research direction.

**Missing References:**

*Do Androids Laugh at Electric Sheep? Humor “Understanding” Benchmarks from The New Yorker Caption Contest* ACL23 best paper

**Paper Topic And Main Contributions:**

The paper addresses the task of "meme captioning" which involves understanding memes and their visual metaphors while interpreting the text associated with the meme. The authors release a new dataset called MemeCap. The authors conduct extensive experiments using state-of-the-art Vision and Language (VL) models to evaluate their performance on the meme captioning task.

**Reasons To Accept:**

This paper introduces an innovative dataset and conducts exhaustive experiments to meticulously evaluate the performance of cutting-edge models, rendering it immensely valuable for future research endeavors.

**Reasons To Reject:**

After reading this paper, it reminded me of one of ACL23's best paper: 'Do Androids Laugh at Electric Sheep? Humor “Understanding” Benchmarks from The New Yorker Caption Contest.' The MemeCap dataset shares a similar concept with this influential ACL paper. However, a notable concern is that this paper fails to cite the aforementioned ACL paper, which is considered unacceptable in scholarly practices. Furthermore, the passage does not delve into the distinctions between these two works, leaving an important aspect unaddressed.

**Reproducibility:**

4: Could mostly reproduce the results, but there may be some variation because of sample variance or minor variations in their interpretation of the protocol or method.

**Reviewer Confidence:**

4: Quite sure. I tried to check the important points carefully. It's unlikely, though conceivable, that I missed something that should affect my ratings.

---

> ### Author Rebuttal · Authors · 2023-08-23
>
> We thank the reviewer for the feedback. We would love to address additional questions during the discussion period if anything is unclear.
>
>
> **Similar to the dataset proposed in "Do Androids Laugh at Electric Sheep? Humor “Understanding” Benchmarks from The New Yorker Caption Contest". No citation of the aforementioned ACL paper**
>
> Thank you for the missing reference. We missed the arXiv version of “Do Androids Laugh at Electric Sheep? Humor ‘Understanding’ Benchmarks from The New Yorker Caption Contest” and the ACL presentation was after the EMNLP submission deadline. We will make sure to cite it in the camera-ready version and elaborate on the similarities and differences between the papers, as we detail here.
>
> First, while both memes and cartoons use a combination of visual and text elements to convey humor, there are major differences between the two. Memes are based on recognizable images that are modified and repurposed to create new meanings based on shared cultural knowledge. Cartoons, on the other hand, are originally drawn illustrations, often in the form of comic strips, that convey a more complex narrative. Second, a more minor difference is that the cartoon dataset focuses on a discriminative matching setup, i.e. selecting the more appropriate caption (as well as generating an explanation), while our meme captioning task is generative, i.e. the task is to generate a caption. It would be interesting for future work that develops a model for one of the tasks, to test how transferable it is to the other; however, we believe that the existence of a cartoon captioning dataset doesn’t affect the significance and usefulness of a meme captioning dataset.
>
> Given that this issue is easy to fix (i.e., we will cite the paper in the camera-ready version), we hope the reviewer will reconsider and not see this as a reason to reject the paper.

---

### Official Review · Reviewer_sFDe · 2023-08-04

**Soundness:** 3

**Excitement:**

4: Strong: This paper deepens the understanding of some phenomenon or lowers the barriers to an existing research direction.

**Paper Topic And Main Contributions:**

The paper proposed a new vision-language dataset of meme pictures and corresponding captions. The scale of the dataset is 6.3K. For each sample, a meme picture, literal image captions, visual metaphors and meme captions are provided. In addition, the authors tested a series of SOTA vision-language and language models on the dataset, showing the lack of meme understanding ability of SOTA models.

**Questions For The Authors:**

A: Why not provide full-training setting on all models?
B: It is interesting to see that Llama, a pure language model, can achieve similar or even better performance than VL models, shown in Table 2. Is there any further explorations on this part?

**Reasons To Accept:**

An novel meme caption dataset to help improve the humorous understanding ability of downstream large models.

**Reasons To Reject:**

There's only the MiniGPT4 model be tested under fine-tuning setting. The other two models, Flamingo and Llama, are not tested under fine-tuning setting.

**Reproducibility:**

4: Could mostly reproduce the results, but there may be some variation because of sample variance or minor variations in their interpretation of the protocol or method.

**Reviewer Confidence:**

4: Quite sure. I tried to check the important points carefully. It's unlikely, though conceivable, that I missed something that should affect my ratings.

---

> ### Author Rebuttal · Authors · 2023-08-23
>
> We thank the reviewer for the detailed feedback.
>
> **Why not provide full-training setting on all models?**
>
> The purpose of the evaluation is to provide baseline results for our new dataset, which is the main contribution of this paper. It was not meant to be comprehensive, but rather to show that the task is interesting and non-trivial. We choose several representative VL models that were publicly available and used them in the standard recommended way. Specifically, Flamingo is supposed to be used in a few-shot setup, as the paper title indicates: https://openreview.net/forum?id=EbMuimAbPbs. We leave it for future work to develop more advanced and/or computationally expensive models for this task.
>
>
> **It is interesting to see that Llama, a pure language model, can achieve similar or even better performance than VL models, shown in Table 2. Is there any further explorations on this part?**
>
> Note that in Table 2, LLaMA always has access to the image caption, which is a (possibly incomplete) textual representation of the image. We hypothesize that in VL models, the vision component lags behind the language component. This claim is supported by prior work [1]. This would make the textual features more useful, which is also evident by the fact that VL models perform better when they are given the OCR text and image caption, which are technically redundant with the image itself. This means that a LM can still perform relatively well without access to visual features. We will clarify this in the camera-ready version.
>
> [1] https://arxiv.org/abs/2102.10407

---

### Official Review · Reviewer_Bzu6 · 2023-08-05

**Soundness:** 3

**Excitement:**

3: Ambivalent: It has merits (e.g., it reports state-of-the-art results, the idea is nice), but there are key weaknesses (e.g., it describes incremental work), and it can significantly benefit from another round of revision. However, I won't object to accepting it if my co-reviewers champion it.

**Missing References:**

1. Do Androids Laugh at Electric Sheep? Humor “Understanding” Benchmarks from The New Yorker Caption Contest, ACL 2023.

**Paper Topic And Main Contributions:**

This work proposes a task of meme captioning and release a new dataset, MEMECAP. The dataset contains memes along with the title, the meme captions, the literal image captions, and the visual metaphors. It utilizes recent Visual-and-Language models to this dataset to analyze their performances on the task of meme captioning.

**Questions For The Authors:**

1. The performance of "MiniGPT4 fine-tuned" is worse than "MiniGPT4 zero-shot". In L437~439, the authors think that it is because "the frozen language and vision model may not have enough information about memes". This claim is not exactly rational. This claim is more proper if the observation is that "MiniGPT4 zero-shot" is bad and "MiniGPT4 fine-tuned" is slightly better than "MiniGPT4 zero-shot".

2. The most important contribution of this work is the dataset with both meme caption and visual metaphor, while there are already existing datasets for visual metaphor. I am still not so clear that whether the task of meme captioning is strongly required, it seems that there is a large overlap on the task of meme captioning and visual metaphor.

3 (Minor, not important). The results/values with "+" in Table 3 (i.e., removing part of input can improve the performance) are not discussed.

**Reasons To Accept:**

1. An interesting dataset with meme, title, literal image caption, visual metaphors, and meme caption.

2. Both automatic evaluation and human evaluations on the task of meme captioning are provided.

**Reasons To Reject:**

1. As a dataset collected by crowdsourcing, there is almost no quality analysis on the collected data (only in the part of human evaluation, using 30 memes, in Section 5.2, the results of "human").

2. There are about 6.3K memes in the datasets, but the authors only utilize 28 memes for the dataset analysis in Section 3.4, and 30 memes for the baseline evaluations in Section 5.2. The numbers of memes used in these evaluations are too small. The evaluations are not convincing enough.

3. The dataset is somewhat similar to (and somewhat different from) the dataset proposed in "Do Androids Laugh at Electric Sheep? Humor “Understanding” Benchmarks from The New Yorker Caption Contest", which gives an explanation (can be regarded as a caption) to a cartoon to understand the humor (potential meaning, similar to metaphor) of the cartoon (similar to meme). Although there are some differences, e.g., one is metaphor and the other is humor, it makes this work less exciting to me.

Other minor comments:

4. In the experiments, this paper uses MiniGPT4 based on LLaMa-13B (L288), but also uses LLaMA-7B (L302). Because the authors want to compare the settings of "without accessing to image, LLaMa" and "with accessing to image, MiniGPT4", it's better to utilize the same model size.

**Reproducibility:**

4: Could mostly reproduce the results, but there may be some variation because of sample variance or minor variations in their interpretation of the protocol or method.

**Reviewer Confidence:**

4: Quite sure. I tried to check the important points carefully. It's unlikely, though conceivable, that I missed something that should affect my ratings.

---

> ### Author Rebuttal · Authors · 2023-08-23
>
> We thank the reviewer for the detailed feedback. We would love to address additional questions during the discussion period if anything is unclear.
>
>
> **Similar to the dataset proposed in "Do Androids Laugh at Electric Sheep? Humor “Understanding” Benchmarks from The New Yorker Caption Contest".**
>
> Thank you for the missing reference. We missed the arXiv version of  “Do Androids Laugh at Electric Sheep? Humor ‘Understanding’ Benchmarks from The New Yorker Caption Contest” and the ACL presentation was after the EMNLP submission deadline. We will make sure to cite it in the camera-ready version and elaborate on the similarities and differences between the papers, as we detail here.
>
> First, while both memes and cartoons use a combination of visual and text elements to convey humor, there are major differences between the two. Memes are based on recognizable images that are modified and repurposed to create new meanings based on shared cultural knowledge. Cartoons, on the other hand, are originally drawn illustrations, often in the form of comic strips, that convey a more complex narrative. Second, a more minor difference is that the cartoon dataset focuses on a discriminative matching setup, i.e. selecting the more appropriate caption (as well as generating an explanation), while our meme captioning task is generative, i.e. the task is to generate a caption. It would be interesting for future work that develops a model for one of the tasks, to test how transferable it is to the other; however, we believe that the existence of a cartoon captioning dataset doesn’t affect the significance and usefulness of a meme captioning dataset.
>
> Given that this issue is easy to fix (i.e., we will cite the paper in the camera-ready version), we hope the reviewer will reconsider and not see this as a reason to reject the paper.
>
> **Small samples for dataset analysis**
>
> Thanks for raising this concern.
>
> Analysis (section 3.4): please note that we manually verified all memes and captions during the data collection phase to guarantee the quality of our dataset. The purpose of the analysis is to provide a more in-depth look into the type of metaphors captured in the data. We have no concerns about the quality of the data. We will clarify this in the paper.
>
> Human evaluation (section 5.2): We chose to evaluate 30 memes based on the convention that a minimum of 30 observations is sufficient to conduct significant statistics. We note that 30 memes x 7 models x 5 evaluation criteria x 3 annotators yield 2,100 annotations in total, which is not trivial in terms of time and cost. Yet, to clear any concerns about the statistical meaningfulness of the evaluation, we will increase the number of evaluated memes for the camera-ready version. We don’t expect the results to change significantly.
>
> **Clarification for "The performance of 'MiniGPT4 fine-tuned' is worse than 'MiniGPT4 zero-shot'" (In L437~439, "the frozen language and vision model may not have enough information about memes")**
>
> Thanks for pointing this out, we were indeed not clear about this. What we meant is that it seems that the zero-shot version generates verbose captions that include a lot of information, often conveying the correct meaning along with irrelevant information such as literal descriptions of the image. The fine-tuned version adapts to the "correct" length but it often fails to focus on the relevant parts, leading to incorrect or incomplete captions. We will clarify this in the camera-ready version.
>
> **Different model size with MiniGPT4 based on LLaMa-13B (L288) and another with LLaMA-7B (L302).**
>
> Thanks for pointing this out. While MiniGPT4 uses the LLaMA-13B model, the OpenFlamingo-9B model is based on LLaMA-7B model. Our initial experiments focused on the OpenFlamingo model considering that MiniGPT4 was released in late April 2023, one month before arxiv deadline, so we chose the LLaMA version accordingly. We will try to include the results from the LLaMA-13B model in the Appendix in the final version of the paper.
>
> **The results/values with "+" in Table 3 (i.e., removing part of input can improve the performance) are not discussed.**
>
> We will include the discussion for the results with “+” in the camera-ready version.

---

### Meta-Review · Area_Chair_QHTr · 2023-09-18

**Recommendation:** 4

**Metareview:**

All reviewers found the dataset to be interesting, innovative, and valuable for future work. Reviewers generally found the experimentation to be thorough, although one reviewer pointed out that a few models were missing fine-tuning results. Two reviewers pointed out a missing citation to an ACL 2023 paper introducing a related (but sufficiently distinct) dataset, which the authors promised to add in the camera ready, addressing these concerns. One reviewer still had remaining concerns about possible data quality, given that the dataset was collected by crowdsourcing and a small number of data points were used in evaluation. However, in my opinion the manual checking of data quality by the authors and the human evaluations are likely sufficient to ensure that the dataset will be useful for future work.

---

### Decision · Program_Chairs · 2023-10-07

**Decision:**

Accept-Main

**Comment:**

All reviewers found the dataset to be interesting, innovative, and valuable for future work. Reviewers generally found the experimentation to be thorough, although one reviewer pointed out that a few models were missing fine-tuning results. Two reviewers pointed out a missing citation to an ACL 2023 paper introducing a related (but sufficiently distinct) dataset, which the authors promised to add in the camera ready, addressing these concerns. One reviewer still had remaining concerns about possible data quality, given that the dataset was collected by crowdsourcing and a small number of data points were used in evaluation. However, in my opinion the manual checking of data quality by the authors and the human evaluations are likely sufficient to ensure that the dataset will be useful for future work.